# Relationship between Alcohol Intake and Chronic Pain with Depressive Symptoms: A Cross-Sectional Analysis of the Shika Study

**DOI:** 10.3390/ijerph19042024

**Published:** 2022-02-11

**Authors:** Takashi Amatsu, Hiromasa Tsujiguchi, Akinori Hara, Sakae Miyagi, Takayuki Kannon, Keita Suzuki, Yukari Shimizu, Thao Thi Thu Nguyen, Kim-Oanh Pham, Fumihiko Suzuki, Tomoko Kasahara, Masaharu Nakamura, Koichiro Hayashi, Aki Shibata, Noriyoshi Ogino, Tadashi Konoshita, Yasuhiro Kambayashi, Hirohito Tsuboi, Atsushi Tajima, Hiroyuki Nakamura

**Affiliations:** 1Department of Hygiene and Public Health, Graduate School of Medical Science, Kanazawa University, 13-1 Takaramachi, Kanazawa 920-8640, Japan; takashi.kamisama@icloud.com (T.A.); t-hiromasa@med.kanazawa-u.ac.jp (H.T.); ahara@m-kanazawa.jp (A.H.); kimoanhpham129@gmail.com (K.-O.P.); kenshoudou@mx3.et.tiki.ne.jp (T.K.); orihciok1003@gmail.com (K.H.); 2Department of Hygiene and Public Health, Faculty of Medicine, Institute of Medical, Pharmaceutical and Health Sciences, Kanazawa University, Kanazawa 920-8640, Japan; keitasuzuk@gmail.com (K.S.); f-suzuki@den.ohu-u.ac.jp (F.S.); m.nakamura.83-7-7@r.vodafone.ne.jp (M.N.); akintoki1116@gmail.com (A.S.); 3Advanced Preventive Medical Sciences Research Center, Kanazawa University, 1-13 Takaramachi, Kanazawa 920-8640, Japan; kannon@med.kanazawa-u.ac.jp (T.K.); atajima@med.kanazawa-u.ac.jp (A.T.); 4Innovative Clinical Research Center, Kanazawa University, 13-1 Takaramachi, Kanazawa 920-8641, Japan; smiyagi@staff.kanazawa-u.ac.jp; 5Department of Bioinformatics and Genomics, Graduate School of Advanced Preventive Medical Sciences, Kanazawa University, 13-1 Takaramachi, Kanazawa 920-8640, Japan; 6Faculty of Health Sciences, Department of Nursing, Komatsu University, 14-1 Mukaimotorimachi, Komatsu 923-0961, Japan; h_zu@me.com; 7Faculty of Public Health, Haiphong University of Medicine and Pharmacy, Ngo Quyen, Hai Phong 180000, Vietnam; nttthao@hpmu.edu.vn; 8Community Medicine Support Dentistry, Ohu University Hospital, Koriyama 963-8611, Japan; 9Department of Environmental Medicine, Faculty of Medicine, Kochi University, Kohasu, Oko-cho, Nankoku City 783-8505, Japan; n-ogino@med.uoeh-u.ac.jp; 10Third Department of Internal Medicine, School of Medicine, University of Occupational and Environmental Health, Iseigaoka 1-1, Yahatanishi-ku, Kitakyushu 807-8555, Japan; 11Third Department of Internal Medicine, University of Fukui Faculty of Medical Sciences, 23-3 Matsuoka Shimoaizuki, Eiheiji-cho, Yoshida-gun, Fukui 910-1193, Japan; konosita@u-fukui.ac.jp; 12Department of Public Health, Faculty of Veterinary Medicine, Okayama University of Science, 1-3 Ikoinooka, Imabari 794-8555, Japan; y-kambayashi@vet.ous.ac.jp; 13Institute of Medical, Pharmaceutical and Health Sciences, Kanazawa University, 1 Kakuma-machi, Kanazawa 920-1192, Japan; tsuboih@p.kanazawa-u.ac.jp

**Keywords:** chronic pain, Geriatric Depression Scale-15 (GDS-15), brief-type self-administered diet history questionnaire (BDHQ), alcohol intake, epidemiology

## Abstract

Although alcohol intake is associated with chronic pain (CP) and analgesia, epidemiological studies have not yet examined the factors affecting the relationship between alcohol intake and CP in detail. Therefore, the present cross-sectional study investigated the relationship between alcohol intake and CP in community-dwelling middle-aged and elderly individuals with/without depressive symptoms. Participants comprised 2223 inhabitants of Shika town in Ishikawa prefecture, located on the Noto Peninsula facing the Sea of Japan, and included 1007 males and 1216 females. CP, depressive symptoms, and alcohol intake were assessed using a CP questionnaire, the Geriatric Depression Scale-15 and the brief-type self-administered diet history questionnaire, respectively. In males without depressive symptoms, mean alcohol intake was significantly higher at 5.70% energy (27.92 g/day) in the CP group than that of 3.75% energy (20.00 g/day) in the non-CP group. The prevalence of low back/knee pain was also significantly higher in males with than in those without depressive symptoms. The present results suggest that long-term alcohol intake is related to CP by reducing the pain threshold and enhancing nociceptive pain as a possible mechanism. However, even a low alcohol intake was associated with psychogenic pain in participants with depressive symptoms. Further studies to investigate the involvement of depressive symptoms and alcohol intake in CP and its prevention are needed.

## 1. Introduction

Chronic pain (CP) is classified as pain that lasts longer than three to six months [1]. The prevalence of CP has been reported to range between 17.5% and 40.2% in Japan [2,3]. Furthermore, 50.2 million adults (20.5%) in the United States have CP [4]. Therefore, a large number of individuals worldwide are affected by CP. CP, which has a negative impact on quality of life [5], has been linked to pre-frailty [3]. Therefore, the primary prevention of CP appears to be important in improving the quality of life and the management of economic losses.

Regarding the relationship between alcohol intake and CP, previous studies have demonstrated that neuropathic pain due to an excessive alcohol intake [6,7], biogenic amines present in alcohol [8], or increased blood serotonin levels after drinking [9] may trigger headaches. However, moderate drinking has been shown to exert analgesic effects [10,11,12]. Therefore, the relationship between alcohol intake and CP has not yet been elucidated in detail. Lasebikan and Gureje [13] conducted a survey on 2149 elderly individuals living in Africa without an alcohol use disorder and showed that an alcohol intake of 7 units (140 g) or more per week increased the prevalence of CP. However, epidemiological findings on the relationship between alcohol intake and CP are inadequate in terms of the duration and amount of alcohol consumed. In our previous epidemiological study on community-dwelling residents [14], we revealed that drinkers with CP had lower serum 25-hydroxyvitamin D levels. We also found an association between inadequate vitamin intake and CP in women in a model adjusted for current drinkers [15]. Therefore, the present study investigated factors that affect the relationship between alcohol intake and CP.

Regarding the relationship between depression and CP, Sheng et al. [16] and Zis et al. [17] reported that CP was more prevalent in individuals with than in those without depression. A review by Sheng et al. [16] indicated that several antidepressants are effective in depression inducing CP treatment. In a cross-sectional study on community-dwelling residents conducted by Lipton [18], depression was less prevalent in mild to moderate drinkers with stress than in those in the extreme drinking category. It has also been shown that females with depression are more likely to consume alcohol than males [19,20]. Therefore, when assessing the relationship between depression and alcohol intake, it is necessary to consider not only the amount of alcohol consumed, but also sex differences.

Although it is known that there is a relationship between higher alcohol intake and CP, it remains unclear what the relationship is when gender-specific depressive symptoms are added to the factor. Therefore, we investigated the relationship between alcohol intake and CP in community-dwelling middle-aged and elderly individuals with/without depressive symptoms by sex.

## 2. Materials and Methods

### 2.1. Participants

We utilized cross-sectional data from the Shika study [14,15,21]. Participants were recruited between October 2013 and December 2016. The target population was residents living in four model districts (Horimatsu, Higashimasuho, Tsuchida, and Togi districts) in Shika town in Ishikawa prefecture, located on the Noto Peninsula facing the Sea of Japan (population, 21,061, population aged 65 years and older on 1 September 2020, 8499 (aging rate 42.2%)) [22]. The major industries in Shika town are electronic component manufacturing, retail, medical care, welfare, followed by agriculture and fishery, which are appropriate fields for various epidemiological studies on a highly aged population. The four districts were randomly selected, and approximately 50% of the population of the town live in these districts. A total of 5013 residents aged 40 years and older live in the model districts. Written informed consent was obtained from all 5013 participants. Of these, 4550 participants underwent a medical examination. Figure 1 shows the participant recruitment chart. In total, 2223 participants (1007 males and 1216 females with mean ages of 68.78 years (SD = 8.41) and 69.65 years (SD = 9.36), respectively) with a reported energy intake of more than 600 kcal and less than 4000 kcal and age range between 40 and 99 years answered the CP questionnaire and Geriatric Depression Scale-15 (GDS-15) and were included in the analysis.

### 2.2. CP

CP was defined as pain that was experienced continuously for more than three months or more than twice a week and had occurred within the last month. We asked participants the following question: ‘‘Do you have pain that has lasted for more than three months?’’, and when they answered “Yes”, they were considered to have CP. We also asked about the location on the body of CP (head, neck, shoulder, upper limb, low back, knee, foot, and others).

### 2.3. Depressive State

We assessed depressive states using the Japanese short version of GDS-15, which consists of 15 questions developed for self-administered surveys [23,24]. Higher scores indicate more severe depressive symptoms [24]. A previous study that evaluated the validity and reliability of the Japanese version of GDS-15 recommended a cut-off score of 6/7 [23]. We used a cut-off score of 5 because we evaluated depressive symptoms rather than depression. We included participants who answered more than 12 out of 15 questions in the analysis.

### 2.4. Alcohol Intake Assessment

Alcohol intake was assessed using the brief type self-administered diet history questionnaire (BDHQ) [25,26]. The BDHQ is a four-page structured questionnaire that assesses the consumption frequency of 58 foods and beverages commonly consumed by the general Japanese population. The BDHQ estimates dietary intake in the last month using an ad hoc computer algorithm. The validity and reliability of the BDHQ has been demonstrated in previous studies [25,26]. Crude data and density methods that estimate intake per 1000 kcal were both used to analyze alcohol intake.

### 2.5. Questionnaire on Demographics

Participants completed a self-administered questionnaire on their socioeconomic status, lifestyle, and medical history including age, sex, duration of education, exercise/hobbies, smoking history, BMI, and medical treatment for diabetes, hyperlipidemia, or hypertension.

### 2.6. Statistical Analysis

Participants were classified into those with/without depressive symptoms. CP was classified into the non-CP and CP groups. IBM SPSS Statistics version 25 for Windows (IBM, Armonk, NY, USA) was used for statistical analyses. The Student’s *t*-test was performed to examine the relationships between continuous variables, while the chi-squared test was used to investigate relationships between categorical variables. A two-way analysis of covariance (ANCOVA) was conducted to examine the main effects and interactions between depressive symptoms and CP on alcohol intake. Adjustments were performed for the following confounding factors: age, living alone, duration of education, exercise/hobbies, smoking history, BMI, and medical treatment for diabetes, hyperlipidemia, or hypertension. A multiple logistic regression analysis was used to examine the effects of depressive symptoms and CP on alcohol intake, using CP at any location on the body as the dependent variable. Independent variables were alcohol intake, age, living alone, duration of education, exercise/hobbies, smoking history, BMI, and medical treatment for diabetes, hyperlipidemia, or hypertension. Analyses were stratified by sex and depressive symptoms. The forced input method was used for variable selection. The significance level was set at 5%.

## 3. Results

### 3.1. Participant Characteristics

Table 1 shows CP, depressive symptoms, and alcohol intake. Among the 2223 participants, there were 1007 males and 1216 females with mean ages of 68.78 years (*SD* = 8.41) and 69.65 years (*SD* = 9.36), respectively. Males were significantly younger than females (*p* < 0.001). The duration of education (*p* = 0.006), alcohol intake (crude data) (*p* < 0.001), and alcohol intake (density method) (*p* < 0.001) were significantly higher in males than in females. Additionally, the proportion of smokers (*p* < 0.001) and those receiving treatment for diabetes (*p* < 0.001) and hypertension (*p* < 0.001) were significantly higher in males than in females. In contrast, the proportion of those living alone (*p* < 0.001), without exercise/hobbies (*p* = 0.031), no alcohol intake (*p* < 0.001), and the rates of any pain (*p* = 0.006) and low back/knee pain (*p* = 0.036) were significantly higher in females than in males.

### 3.2. Comparisons between Participants with/without Depressive Symptoms

Table 2 shows a comparison of participants with/without depressive symptoms in males. No significant differences were observed in mean age between the 657 participants without depressive symptoms (68.55 years) and the 350 participants with depressive symptoms (69.21 years, *p* = 0.260). The proportion of participants living alone (*p* = 0.003), without exercise/habits (*p* < 0.001), pain (*p* = 0.019), and low back/knee pain (*p* = 0.003) were significantly higher in participants with than in those without depressive symptoms. No significant differences were observed in alcohol intake (density method) between the two groups.

Among females, mean age was significantly younger in the 820 participants without depressive symptoms (68.80 years) than in the 396 participants with depressive symptoms (71.39 years, *p* < 0.001) (Appendix A). The proportion of females without exercise/habits (*p* < 0.001), any pain (*p* < 0.001), low back/knee pain (*p* < 0.001), and other pain was significantly higher in participants with than in those without depressive symptoms. In contrast, the duration of education (*p* = 0.025), BMI (*p* = 0.040), and proportion receiving treatment for hyperlipidemia (*p* < 0.001) were significantly higher in participants without than in those with depressive symptoms. No significant differences were observed in alcohol intake (density method) between the two groups.

### 3.3. Comparison between CP Groups

Table 3 shows a comparison of the two CP groups in males. Mean age was significantly younger in the 933 participants in the non-CP group (68.60 years) than in the 74 participants in the CP group (71.03 years, *p* = 0.004). The proportion of males living alone (*p* = 0.039) and with depressive symptoms (*p* = 0.019) was significantly higher in the CP group than in the non-CP group. Furthermore, the duration of education (*p* < 0.001) was significantly higher in the non-CP group than in the CP group. No significant differences were observed in alcohol intake between the two groups.

Among the females, mean age was significantly younger in the 1085 participants in the non-CP group (69.23 years) than in the 131 participants in the CP group (73.08 years, *p* < 0.001) (Appendix A). The proportion of females with depressive symptoms (*p* < 0.001) was significantly higher in the CP group than in the non-CP group. In contrast, the duration of education (*p* = 0.002) and the proportion receiving treatment for hyperlipidemia (*p* = 0.002) were significantly lower in the CP group than in the non-CP group. No significant differences were observed in alcohol intake (density method) between the two groups.

### 3.4. Effects of the Interaction between Depressive Symptoms and CP Groups

Table 4 shows the main effects and interactions between depressive symptoms and CP on alcohol intake stratified by the location of pain in males. A two-way ANCOVA was used to analyze the main effects and interactions in participants with and without depressive symptoms and in the two CP (any) groups. The dependent variable was alcohol intake (density method), and confounders were adjusted for age, living alone, duration of education, without exercise/hobbies, smoking history, BMI, diabetes treatment, hyperlipidemia treatment, and hypertension treatment. The main effect was shown for alcohol intake (density method) in CP (any) (*p* = 0.002) and CP (low back/knee) (*p* = 0.043) between the two depressive symptoms groups. In addition, the interaction was observed for alcohol intake (density method) in CP (any) between the two depressive symptoms groups and the two CP groups (*p* = 0.025). These results suggest that alcohol intake (density method) was significantly higher in participants without than in those with depressive symptoms in the CP group. In contrast, alcohol intake was similar between participants with and without depressive symptoms in the non-CP group.

Among females, alcohol intake (density method) associated with CP at any location in the body showed no main effect or interaction (Appendix A).

### 3.5. Multiple Logistic Regression Analysis of the Relationship between Alcohol Intake and CP Stratified by Depressive Symptoms

Table 5 shows the results of a multiple logistic regression analysis of the relationship between alcohol intake and CP stratified by depressive symptoms in males. Alcohol intake (density method) was a significant independent variable for CP (any) in participants without depressive symptoms (OR: 1.090; 95%CI: 1.018, 1.167; *p* = 0.013), but not in those with depressive symptoms after adjustments for confounding factors. Alternatively, among females, alcohol intake (density method) was not a significant independent variable in either group (Appendix A). These results suggest that a high alcohol intake in participants without depressive symptoms was associated with CP in males only.

## 4. Discussion

The main result of the present study was that mean alcohol intake in males without depressive symptoms was significantly higher at 5.70% energy (27.92 g/day) in the CP group than at 3.75% energy (20.00 g/day) in the non-CP group. Furthermore, the prevalence of low back/knee pain was significantly higher in males with than in those without depressive symptoms.

The pathophysiological mechanism of CP in individuals without depressive symptoms has been explained as neuropathic pain, which is triggered by a long-term alcohol intake, and subsequently damages peripheral nerves [27,28] or increases nociceptive pain [29,30]. Regarding the relationship between alcohol intake and pain, an intolerance of histamine, which is one of the biogenic amines present in alcohol such as wine induces headaches [8]. A mini-review by Panconesi [31] revealed that alcohol itself triggers headaches, particularly when ingested at large quantities, and some components of alcoholic drinks may reinforce the effects of alcohol or vice versa. An experimental study by Boyer et al. [9] showed that some alcoholic beverages increased blood serotonin levels. In animal studies with rats conducted by Zhang et al. [32], serotonin released from dural mast cells appeared to promote headaches by sensitizing intracranial meningeal nociceptors. Since mast cells are present throughout the body, they are considered to be involved in CP other than headaches. The mechanisms by which pain occurs during the process of alcohol decomposition after its intake have been investigated. Experiments with rats performed by Fu et al. [33] revealed increased mechanical and thermal sensitivities with chronic and intermittent drinking over 12 h to seven days or longer after desorption. In contrast, a moderate alcohol intake has been reported to exert analgesic effects. A randomized, double-blind, and placebo-controlled intervention study on healthy adults by Capito et al. [10] demonstrated that low doses of alcohol were particularly effective for analgesia. As an underlying mechanism for the analgesic effects of alcohol intake, Lovinger and Roberto [11] considered a decrease in excitatory neurotransmission due to the rapid inhibition of the ionotropic glutamate receptor activity of alcohol. Lobo and Harris [12] reported that the inhibitory effects of gamma-aminobutyric acid were associated with hypoalgesia following acute alcohol intake. The present results showed that mean alcohol intake by participants without depressive symptoms in the CP group was 27.92 g/day, which was approximately 700 mL per day when calculated by the amount of beer. Lasebikan and Gurejet [13] previously revealed that alcohol intake associated with CP was greater than 20 g per day. Taken together with the present results showing an alcohol intake of 20 g/day in the non-CP group, this amount of alcohol is not sufficient to cause CP. Few studies have investigated the epidemiological relationship between alcohol intake and CP [13] based on a detailed nutritional survey of average intake in one month. Therefore, the present results provide supportive evidence for alcohol intake being a cause of the onset of CP because long-term alcohol intake is considered to be associated with the onset of CP, even if it is not excessive.

Regarding the relationship between depressive symptoms and CP, the prevalence of low back/knee pain was significantly higher in males with than in those without depressive symptoms. Psychogenic pain has been implicated in depression and CP [34,35,36]. A study on quantitative sensory tests conducted by depressed patients suggested central hyperexcitability due to decreased serotonin production as a mechanism for CP due to a reduction in the cold pain threshold and an increased responsiveness to repetitive noxious mechanical stimuli [37]. Although the exact mechanisms have not yet been elucidated, a review by Ossipov et al. [38] indicated that serotonin/noradrenaline reuptake inhibitors and other drugs that act on opioid-sensitive circuits control CP associated with depression by activating the descending circuit at the spinal cord level. In an epidemiological study that investigated pain sites and psychological conditions, Picavet et al. [39] suggested that a relationship exists between chronic low back pain and catastrophic thinking. Somers et al. [40] also revealed that pain was associated with psychological distress in patients with knee osteoarthritis. The present results, showing a relationship between depressive symptoms and low back/knee pain, appear to support the mechanism in which depression impairs the descending pain inhibitory system, resulting in CP in load-bearing joints such as the trunk and lower limbs.

Our results also revealed that alcohol intake by males with depressive symptoms did not significantly differ between the non-CP and CP groups. A cohort study by Maleki et al. [41] reported a relationship between depression and CP with or without alcohol abuse. The novel result of the present study is that depression and CP were associated with a low alcohol intake. The administration of medication and physical therapy for analgesia are conducted as treatments for CP. However, from a preventive medical viewpoint, guidance on nociceptive or neuropathic pain needs to focus on drinking habits and that on psychogenic pain on improvements in the depressive status in males. Since mean alcohol intake by females was as low as 0.51% energy, a comparison with that by males was not possible.

The strength of the present study is that an almost-exhaustive epidemiological survey was performed. In addition, by using two methods of analysis, two-way ANCOVA and multiple logistic regression analysis, we were able to elucidate in detail the relationship between alcohol intake, depressive symptoms, and CP. However, the limitation of this study, which was cross-sectional in nature, was that it was not possible to elucidate the causal relationship between alcohol intake, CP, and depressive symptoms. Second, the self-administration of BDHQ, GDS, and CP may lack objectivity. Third, because years have passed since the survey, the results of our analysis might be inconsistent with the current situation. Fourth, since the survey was conducted in one region of Japan, the results may differ from those of other countries. Finally, we did not include the relationship between nutrients and CP in the analysis.

## 5. Conclusions

The Shika study on community-dwelling middle-aged and elderly individuals showed a significantly higher alcohol intake by males without depressive symptoms in the CP group than in the non-CP group, indicating that low back/knee pain was significantly more prevalent in males with than in those without depressive symptoms. Collectively, the present results suggest that long-term alcohol intake is related to CP in participants without depressive symptoms due to a lowering of the pain threshold and enhancements in nociceptive pain as a possible mechanism. On the other hand, the present study showed that even a low alcohol intake was associated with psychogenic pain in participants with depressive symptoms. Therefore, depression and alcohol intake are important factors to consider in the prevention of CP.

## Figures and Tables

**Figure 1 ijerph-19-02024-f001:**
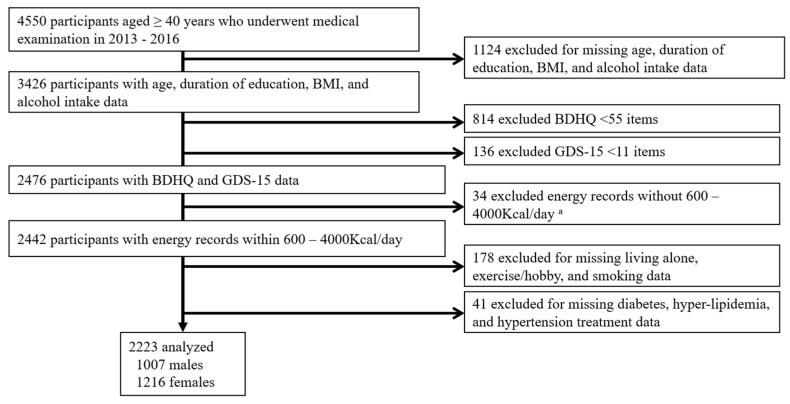
Participant recruitment chart. ^a^ This reference value was chosen for the following reasons: less than 600 kcal/day is equivalent to half the energy intake required for the lowest physical activity category; more than 4000 kcal/day is equivalent to 1.5 times the energy intake required for the medium physical activity category. Abbreviations: BMI, body mass index; BDHQ, brief-type self-administered diet history questionnaire; GDS, the Geriatric Depression Scale.

**Table 1 ijerph-19-02024-t001:** Participant characteristics.

	Total (*n* = 2223)				*p*-Value ^a^
	Male (*n* = 1007)		Female (*n* = 1216)		
	Mean/*n*	*SD*/%	Mean/*n*	*SD*/%	
Age, years	68.78	8.41	69.65	9.36	**<0.001**
Living alone, *n*	75	7.45	150	12.34	**<0.001**
Education, years	11.39	3.01	10.88	2.50	**0.006**
Without exercise/hobbies, *n*	601	59.68	780	64.14	**0.031**
Smoking history, *n*	284	28.20	44	3.62	**<0.001**
BMI, kg/m^2^	23.41	2.97	22.64	3.23	0.079
Diabetes treatment, *n*	131	13.01	93	7.65	**<0.001**
Hyperlipidemia treatment, *n*	111	11.02	218	17.93	**<0.001**
Hypertension treatment, *n*	363	36.05	372	30.59	**0.007**
Depressive symptoms, *n*	350	34.76	396	32.57	0.276
Alcohol (crude data), g	19.85	25.09	2.02	6.30	**<0.001**
Alcohol (density method), % energy	3.80	4.53	0.51	1.54	**<0.001**
No alcohol intake, *n*	325	32.27	908	74.67	**<0.001**
CP					
Any, *n*	74	7.35	131	10.77	**0.006**
Head, *n*	0	–	4	0.329	0.089
Neck/shoulder/upper limb, *n*	28	2.78	46	3.78	0.190
Low back/knee, *n*	56	5.56	95	7.81	**0.036**
Foot, *n*	15	1.49	23	1.89	0.467

Notes: ^a^ *p*-values were calculated using the Student’s *t*-test for continuous variables and the chi-squared test for categorical variables (*p*-values less than 0.05 are highlighted in bold). Abbreviations: SD, standard deviation; BMI, body mass index; CP, chronic pain.

**Table 2 ijerph-19-02024-t002:** Differences between the participants with and without depressive symptoms in males.

	Without Depressive Symptoms (*n* = 657)		With Depressive Symptoms (*n* = 350)		*p*-Value ^a^
	Mean/*n*	*SD*/%	Mean/*n*	*SD*/%	
Age, years	68.55	7.84	69.21	9.38	0.260
Living alone, *n*	37	5.63	38	10.86	**0.003**
Education, years	11.49	3.23	11.21	2.56	0.130
Without exercise/hobbies, *n*	347	52.82	254	72.56	**<0.001**
Smoking history, *n*	184	28.01	100	28.57	0.849
BMI, kg/m^2^	23.53	2.84	23.19	3.18	0.089
Diabetes treatment, *n*	87	13.24	44	12.57	0.763
Hyperlipidemia treatment, *n*	81	12.33	30	8.57	0.070
Hypertension treatment, *n*	248	37.75	115	32.86	0.124
Alcohol (crude data), g	20.89	24.84	17.90	25.47	0.075
Alcohol (density method), % energy	3.95	4.38	3.53	4.79	0.175
No alcohol intake, *n*	190	28.92	135	38.57	**0.002**
CP					
Any, *n*	39	5.94	35	10.00	**0.019**
Head, *n*	0	-	0	-	-
Neck/shoulder/upper limb, *n*	15	2.28	13	3.71	0.188
Low back/knee, *n*	27	4.11	29	8.29	**0.006**
Foot, *n*	7	1.07	8	2.29	0.128

Notes: ^a^ *p*-values were calculated using the Student’s *t*-test for continuous variables and the chi-squared test for categorical variables (*p*-values less than 0.05 are highlighted in bold). Abbreviations: SD, standard deviation; BMI, body mass index; CP, chronic pain.

**Table 3 ijerph-19-02024-t003:** Differences between the chronic pain groups in males.

	Non-CP (Any) (*n* = 933)		CP (Any) (*n* = 74)		*p*-Value ^a^
	Mean/*n*	*SD*/%	Mean/*n*	*SD*/%	
Male (*n* = 1007)					
Age, years	68.60	8.28	71.03	9.73	**0.040**
Living alone, *n*	65	6.97	10	13.51	**0.039**
Education, years	11.48	3.03	10.27	2.56	**<0.001**
Without exercise/hobbies, *n*	553	59.27	48	64.86	0.345
Smoking history, *n*	258	27.65	26	35.14	0.169
BMI, kg/m^2^	23.36	2.96	24.02	3.03	0.076
Diabetes treatment, *n*	119	12.75	12	16.22	0.394
Hyperlipidemia treatment, *n*	103	11.04	8	10.81	0.952
Hypertension treatment, *n*	336	36.01	27	36.49	0.935
Depressive symptoms, n	315	33.76	35	47.30	**0.019**
Alcohol (crude data), g	19.73	25.04	21.39	25.76	0.594
Alcohol (density method), % energy	3.75	4.44	4.38	5.52	0.347
No alcohol intake, *n*	302	32.37	23	31.08	0.820
CP					
Head, *n*	0	-	0	-	-
Neck/shoulder/upper limb, *n*	0	-	28	37.84	-
Low back/knee, *n*	0	-	56	75.68	-
Foot, *n*	0	-	15	20.27	-

Notes: ^a^ *p*-values were calculated using the Student’s *t*-test for continuous variables and the chi-squared test for categorical variables (*p*-values less than 0.05 are highlighted in bold). Abbreviations: SD, standard deviation; BMI, body mass index; CP, chronic pain.

**Table 4 ijerph-19-02024-t004:** Interactions between depressive symptoms and CP on alcohol intake (density method) in males.

		Non-CP (Any) (*n* = 933)		CP (Any) (*n* = 74)		*p*-Value ^a^		
		Mean (95%CI)	*n*	Mean (95%CI)	*n*	DS	CP	DS × CP
Head pain	NDS	3.87 (3.53, 4.21)	657	- (-, -)	0	0.503	-	-
	DS	3.67 (3.20, 4.14)	350	- (-, -)	0			
Neck/shoulder/ upper limb pain	NDS	3.85 (3.51, 4.20)	642	4.59 (2.37, 6.81)	15	0.727	0.448	0.912
	DS	3.65 (3.18, 4.13)	337	4.20 (1.81, 6.59)	13			
Low back/knee pain	NDS	3.81 (3.47, 4.16)	630	5.18 (3.51, 6.86)	27	**0.043**	0.75	0.053
	DS	3.75 (3.27, 4.24)	321	2.77 (1.17, 4.37)	29			
Foot pain	NDS	3.85 (3.51, 4.19)	650	5.43 (2.16, 8.69)	7	0.102	0.886	0.129
	DS	3.72 (3.24, 4.19)	342	1.81 (−1.23, 4.86)	8			

Notes: ^a^ Analysis of covariance (*p*-values less than 0.05 are highlighted in bold). Adjusted for BMI, age, living alone, education, without exercise/hobbies, smoking history, BMI, diabetes treatment, hyperlipidemia treatment, and hypertension treatment. Abbreviations: CP, chronic pain; NDS, no depressive symptoms; DS, depressive symptoms; CI, confidence interval; BMI, body mass index.

**Table 5 ijerph-19-02024-t005:** Relationship between alcohol intake and CP stratified by sex and depressive symptoms.

		Exp (B)	95%Cl Lower	96%Cl Upper	*p*-Value
With depressive symptoms (*n* = 350)	Alcohol (density method)	0.986	0.905	1.074	0.746
Without depressive symptoms (*n* = 657)	Alcohol (density method)	1.090	1.018	1.167	**0.013**

Notes: Significant estimates are in bold. Adjusted for BMI, age, living alone, education, without exercise/hobbies, smoking history, BMI, diabetes treatment, hyperlipidemia treatment, and hypertension treatment. Abbreviations: Exp (B), Exponentiation of the B coefficient; CI, confidence interval.

## Data Availability

Data in the present study are available upon request from the corresponding author. Data are not publicly available due to privacy and ethical policies.

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
