# Peer review of "Relationship between Alcohol Intake and Chronic Pain with Depressive Symptoms: A Cross-Sectional Analysis of the Shika Study"

_ijerph, 2022, doi:10.3390/ijerph19042024_

Round 1
Reviewer 1 Report
The study's aim was to analyse the relationship between chronic pain and alcohol intake in more than 2,000 middle-aged and elderly individuals with or without depressive symptoms.
Overall, the manuscript is well-prepared and includes all crucial information.
The subject of the paper is very interesting. However, looking at chronic pain through the lens of alcohol consumption alone seems one-sided. The limitation of this study is the lack of analysis of diet quality of respondents, such as dietary patterns containing anti-inflammatory ingredients. Inflammation and oxidative stress are the main pathophysiological pathways of chronic pain. Therefore, additional analysis related to dietary quality should be added to the manuscript and discussed in the context of other research.
Some small corrections are also required in the tables:
- Table 1. The entire cohort of men and women is listed as 2223 but the individual cohorts (men: n = 1429; women: n = 1733) actually add up to 3162.
- Table 2. The number of men without (n = 667) and with depressive symptoms (n = 350) does not add up to total (n = 1007); the table heading does not include this information for women.
- Tables 3-4. The table heading does not include information for the number of women in subgroups (non-chronic pain; chronic pain).
Author Response
Comment to reviewer 1
Comments and Suggestions for Authors
The study's aim was to analyse the relationship between chronic pain and alcohol intake in more than 2,000 middle-aged and elderly individuals with or without depressive symptoms.
Overall, the manuscript is well-prepared and includes all crucial information.
Point 1
The subject of the paper is very interesting. However, looking at chronic pain through the lens of alcohol consumption alone seems one-sided. The limitation of this study is the lack of analysis of diet quality of respondents, such as dietary patterns containing anti-inflammatory ingredients. Inflammation and oxidative stress are the main pathophysiological pathways of chronic pain. Therefore, additional analysis related to dietary quality should be added to the manuscript and discussed in the context of other research.
Response 1
Thanks for your proper advice.
We have already published an article on the association between nutrients and chronic pain, with the covariates adjusted for current drinkers. Therefore, we have added to the introduction section of the revised manuscript as follows: "We also found an association between inadequate vitamin intake and CP in women in a model adjusted for current drinkers [15]". (P2 L82-83)
Therefore, we did not mention the relationship between nutrient intake and chronic pain in the present article because it overlaps with our previous study.
However, we have added to the limitations section of the revised manuscript as follows: "Finally, we did not include the relationship between nutrients and CP in the analysis. (P10 L485-486)
Point 2
Some small corrections are also required in the tables:
Table 1. The entire cohort of men and women is listed as 2223 but the individual cohorts (men: n = 1429; women: n = 1733) actually add up to 3162.
Response 2
In Table 1 of the revised manuscript, the number of males was corrected to 1007 and that of females to 1216. (P5 L239)
Point 3
Table 2. The number of men without (n = 667) and with depressive symptoms (n = 350) does not add up to total (n = 1007); the table heading does not include this information for women.
Tables 3-4. The table heading does not include information for the number of women in subgroups (non-chronic pain; chronic pain).
Response 3
Tables 2, 3, 4, and 5 in the revised manuscript have been changed to include only males for easier understanding. The number of participants in each group has been changed accordingly. The female portions of the tables are listed in supplementary materials as Tables S1, S2, S3, and S4.

Reviewer 2 Report
First of all, I would like to thank the journal and its editor for the possibility of reviewing this manuscript entitled "Relationship between alcohol intake and depressive symptoms with chronic pain: A cross-sectional analysis of the Shika study" that opts to be published in the journal. IJERPH.
The manuscript deals with a current topic, such as alcohol consumption, depressive symptoms and chronic pain, in middle-aged and elderly subjects. A higher percentage is observed in men and, in addition, with a greater relationship with subjects who do not have depressive symptoms. With this result, it is possible to emphasize these variables individually and collectively.
The introduction is well founded and ends with the objective of the study, which is very important to fully enter into the methodology.
The sample is broad, homogeneous between men and women, but in my opinion the sample is obsolete, 6 years have already passed... in which there have been many changes, including a pandemic that has surely affected these results.
In the abstract, on line 46, Shika town appears, it should be explained where it is. As in the methodology, in addition to why this geographical area has been selected.
The discussion begins with some results. I think the first paragraph does not correspond to this section (lines 265-269).
Author Response
Comment to reviewer 2
Comments and Suggestions for Authors
First of all, I would like to thank the journal and its editor for the possibility of reviewing this manuscript entitled "Relationship between alcohol intake and depressive symptoms with chronic pain: A cross-sectional analysis of the Shika study" that opts to be published in the journal. IJERPH.
The manuscript deals with a current topic, such as alcohol consumption, depressive symptoms and chronic pain, in middle-aged and elderly subjects. A higher percentage is observed in men and, in addition, with a greater relationship with subjects who do not have depressive symptoms. With this result, it is possible to emphasize these variables individually and collectively.
The introduction is well founded and ends with the objective of the study, which is very important to fully enter into the methodology.
Point 1
The sample is broad, homogeneous between men and women, but in my opinion the sample is obsolete, 6 years have already passed... in which there have been many changes, including a pandemic that has surely affected these results.
Response 1
As Shika Town in Ishikawa Prefecture is a community with a highly aged population, in addition to less migration, we do not expect the population structure to change rapidly in 6 years. Furthermore, please understand that we aimed to investigate the relationship between alcohol intake, depressive symptoms, and chronic pain under normal circumstances, not to investigate the effects of a covid-19 pandemic.
Point 2
In the abstract, on line 46, Shika town appears, it should be explained where it is. As in the methodology, in addition to why this geographical area has been selected.
Response 2
We have added to the abstract and method section of the revised manuscript as follows: "Shika town in Ishikawa prefecture, located on the Noto Peninsula facing the Sea of Japan". (P1 45, P2 L104)
In addition, we have added to the method section of the revised manuscript as follows: “The major industries in Shika town are electronic component manufacturing, retail, medical care, welfare, followed by agriculture and fishery, which are appropriate fields for various epidemiological studies on a highly aged population”. (P3 L161-163)
Point 3
The discussion begins with some results. I think the first paragraph does not correspond to this section (lines 265-269).
Response 3
The relevant statement for Table 1 in the discussion section is as follows: "Since mean alcohol intake by females was as low as 0.51% energy, a comparison with that by males was not possible". (P9 L446-P10 L475)

Reviewer 3 Report
I offer the following comments.
- Please re-examine the title. I think you looked at the relationship among alcohol intake, depression symptoms and chronic pain. This consideration may affect the purpose in lines 41-42.
- Line 46 – why was nutrition included? Line 50 – I do not believe you can state something caused chronic pain when this was a relationship study.
- In the introduction, at least two additional variables are added – sex and age. How were they integrated into the purpose?
- Data for this study were collected from 2013-2016. This was up to 9 years ago. Did any cultural/historical changes affect the findings since data are old?
- What were the research questions for the study? These questions will focus your literature and the variables for the study.
- IRB was acknowledged.
- How did reported energy fit the study (line 104)? Nutritional survey? It appears that you keep adding variables.
- For all instruments, please state at least reliability. It is not sufficient to say it can be found elsewhere. Also state the reliability of instruments using your data. How long did it take to complete the instrument? If more than one instrument was used for a variable, how were scores combined? You included so many variables, it is as if you were “fishing” or just looking for whatever variable(s) would be significant.
- You present multiple pages of varied statistics. For example, Table 2 is over a page long. Why so many variables? What is a more effective way to analyze? I encourage consultation with a statistician as to the most effective ways to analyze your data. If what you did is most effective, please write a paragraph about why it is so. I highly encourage research questions or hypotheses.
- You related some findings to the literature. What are other strengths versus the one you list? The limitations must be described more thoroughly. For example, your data are old. You did not mention culture.
- The conclusions do not bring in the multiple variables studied.
- There are 42 references but only around 8 from the last 5 years. Please explore if there are more recent references.
- The tables are ok but please consider shortening once you decide upon critical variables.
Author Response
Comment to reviewer 3
Comments and Suggestions for Authors
I offer the following comments.
Point 1
Please re-examine the title. I think you looked at the relationship among alcohol intake, depression symptoms and chronic pain. This consideration may affect the purpose in lines 41-42.
Response 1
We have amended the title of the revised manuscript as follows: " Relationship between alcohol intake and chronic pain with depressive symptoms: A cross-sectional analysis of the Shika study". (P1 L2-4)
Point 2
Line 46 – why was nutrition included?
Response 2
Although crude data on alcohol drinking alone can assess the amount of alcohol consumed per day, it cannot evaluate the ratio of alcohol consumption to daily caloric intake. The brief-type self-administered diet history questionnaire uses a computer algorithm to assess the amount of food and drink per calorie intake using the density method (% energy), which we utilized for our main results.
Point 3
Line 50 – I do not believe you can state something caused chronic pain when this was a relationship study.
Response 3
In the abstract section of the revised manuscript, the word "caused" has been amended as follows: "The present results suggest that long-term alcohol intake is related to CP by reducing the pain threshold and enhancing nociceptive pain as a possible mechanism". (P1 L51-P2 L56)
Point 4
In the introduction, at least two additional variables are added – sex and age. How were they integrated into the purpose?
Response 4
We have changed to the introduction section of the revised manuscript as follows: “we investigated the relationship between alcohol intake and CP in community-dwelling middle-aged and elderly individuals with/without depressive symptoms by sex”. (P2 L96-98)
Although depressive symptoms and chronic pain are targeted in the cited papers because they are more common in middle-aged and older people, we included age as a covariate. Please understand that we do not perform analyses categorized by age.
Point 5
Data for this study were collected from 2013-2016. This was up to 9 years ago. Did any cultural/historical changes affect the findings since data are old?
Response 5
As Shika Town in Ishikawa Prefecture is a community with a highly aged population, in addition to less migration, we do not expect the population structure to change rapidly in 6 years. Furthermore, please understand that we aimed to investigate the relationship between alcohol intake, depressive symptoms, and chronic pain under normal circumstances, not to investigate the effects of a covid-19 pandemic.
Point 6
What were the research questions for the study? These questions will focus your literature and the variables for the study.
Response 6
We have added the research question and changed to the introduction section of the revised manuscript as follows: “Although it is known that there is a relationship between higher alcohol intake and CP, it remains unclear what the relationship is when gender-specific depressive symptoms are added to the factor. Therefore, we investigated the relationship between alcohol intake and CP in community-dwelling middle-aged and elderly individuals with/without -depressive symptoms by sex”. (P2 L94-98)
Point 7
IRB was acknowledged.
Point 8
How did reported energy fit the study (line 104)? Nutritional survey? It appears that you keep adding variables.
Response 8
As we responded in Response 2, we use the density method to measure the percentage of alcohol intake per daily calorie consumed. Hence, we exclude participants who are extremely hypo- and over-nourished, as explained in Figure 1.
Point 9
For all instruments, please state at least reliability. It is not sufficient to say it can be found elsewhere. Also state the reliability of instruments using your data. How long did it take to complete the instrument? If more than one instrument was used for a variable, how were scores combined?
Response 9
Regarding the reliability of GDS-15, we have amended to the methods section of the revised manuscript as follows: “A previous study that evaluated the validity and reliability of the Japanese version of GDS-15 recommended a cut-off score of 6/7 [23]”. (P3 L185-187)
Regarding the reliability of BDHQ, we have amended to the methods section of the revised manuscript as follows: “The validity and reliability of the BDHQ has been demonstrated in previous studies [25, 26]”. (P3 L196-197)
Point 10
You included so many variables, it is as if you were “fishing” or just looking for whatever variable(s) would be significant. You present multiple pages of varied statistics. For example, Table 2 is over a page long. Why so many variables? What is a more effective way to analyze? I encourage consultation with a statistician as to the most effective ways to analyze your data. If what you did is most effective, please write a paragraph about why it is so. I highly encourage research questions or hypotheses.
Response 10
We have removed a few items from the tables. Tables 2, 3, 4, and 5 in the revised manuscript have been changed to include only males for easier understanding. The female portions of the tables are listed in supplementary materials as Tables S1, S2, S3, and S4. In Table 5, we have included the covariates in the table explanation, rather than presenting them as variables.
Pont 11
You related some findings to the literature. What are other strengths versus the one you list? The limitations must be described more thoroughly. For example, your data are old. You did not mention culture.
The conclusions do not bring in the multiple variables studied.
Response 11
We have added to the discussion section of the revised manuscript as follows: “In addition, by using two methods of analysis, two-way ANCOVA and multiple logistic regression analysis, we were able to elucidate in detail the relationship between alcohol intake, depressive symptoms, and CP”. (P10 L477- 479)
We have also added to the discussion section of the revised manuscript as follows: “Thirdly, because years have passed since the survey, the results of our analysis might be inconsistent with the current situation. Fourthly, since the survey was conducted in one region of Japan, the results may differ from those of other countries. Finally, we did not include the relationship between nutrients and CP in the analysis”. (P10 L482-486)
Point 12
There are 42 references but only around 8 from the last 5 years. Please explore if there are more recent references.
Response 12
We have removed several references from the revised manuscript. In addition, we have corrected reference numbers 1, 3, 4, 5, 7, 15, 16, 17, 19, 20, 24, 26, 29, and 30 to be within 5 years.
Point 13
The tables are ok but please consider shortening once you decide upon critical variables.
Response 13
Same as response 10.